# Genetic Parameters of Workability Traits in the Population of Polish Holstein-Friesian Cows Based on Conventional and Genomic Data

**DOI:** 10.3390/ani11082443

**Published:** 2021-08-19

**Authors:** Bartosz Szymik, Piotr Topolski, Wojciech Jagusiak

**Affiliations:** 1Department of Cattle Breeding, The National Research Institute of Animal Production, 2, Sarego Street, 31-047 Kraków, Poland; piotr.topolski@iz.edu.pl; 2Department of Genetics and Animal Breeding, Faculty of Animal Science, University of Agriculture in Kraków, Al. Mickiewicza 24/28, 31-059 Kraków, Poland; rzjagusi@cyfronet.pl

**Keywords:** dairy cattle, milking speed, milking temperament, heritabilities, genetic and phenotypic correlations, genomic selection

## Abstract

**Simple Summary:**

Workability traits are a group of functional traits that affect the economics of dairy production and are increasingly included in selection indexes. The most important of them include milking speed and temperament. The aim of this study was to estimate genetic and phenotypic parameters of workability traits. The estimation was carried out by considering two approaches: the first using pedigree data and the second using pedigree and genomic data. The obtained results indicate that workability traits belong to low heritable traits and are positively correlated genetically and phenotypically, which means a possibility of their effective improvement in the population, taking into account that the genomic information of sires did not have a significant effect on the estimated genetic parameters.

**Abstract:**

Heritabilities of workability (WT) traits—milking speed (MS) and temperament (MT)—as well as genetic and phenotypic correlations between these traits in the population of Polish Holstein-Friesian (PHF) cows were estimated. The estimation of genetic parameters was performed twice: first with the use of pedigree data; and second with the use of pedigree and genomic data. Phenotypic data from routinely conducted MS and MT evaluations for 1,045,511 cows born from 2004 to 2013 were available; the cows were evaluated from 2011 to 2015. The main dataset was reduced based on imposed restrictions (e.g., on age of calving, stage of lactation and day of first trial milking). The dataset prepared in this manner comprised 391,615 cows. It was then reduced to daughters of 10% randomly selected sires for computational reasons. Finally, for genetic parameter estimation, 13,280 records of cows were used. The linear observation model included additive random effects of animal, fixed effects of herd-year-season of calving subclass (HYS) and lactation phase, fixed regressions on cow age at calving and the percent of HF breed genes in the cow genotype. Heritabilities estimated based on pedigree data were 0.12 (±0.0067) for MS and 0.08 (±0.0063) for MT, the genetic correlation between MS and MT was estimated at 0.05 (±0.0002) and the phenotypic correlation coefficient was estimated at 0.14 (±0.0004). The inclusion of genomic information of sire bulls had no clear effect on the size of the estimated WT genetic parameters. The heritabilities of MS and MT were 0.11 (±0.0065) and 0.09 (±0.0012), respectively. The genetic and phenotypic correlation coefficients were 0.07 (±0.0003) and 0.12 (±0.0005), respectively. The sizes of the obtained heritabilities of WT and of the genetic and phenotypic correlation between these traits indicate the possibility of effective population improvement for both WT traits.

## 1. Introduction

In a review of selection indexes in countries with developed cattle breeding, Miglior et al. [1] found that the importance of functional traits in dairy cattle breeding has increased significantly since the early 2000s. The group of functional traits, which includes workability (WT) traits, are traits that are not directly related to milk yields but affect the profitability of milk production by reducing its costs. In recent years, the importance of WT, such as milking speed (MS) and temperament (MT), has been increasing in cattle breeding programmes. MS can be defined as the cow’s ability to milk in a short time, while MT is the cow’s behaviour and ease of handling during milking [2,3]. The main aim of estimating the breeding value of WT is to identify sire bulls from which daughters are born with undesirable phenotype in terms of MS or MT [4]. Several studies have shown that WTs were a determining factor on culling cows from the herd [5,6,7,8]. WTs are considered as a very important group of traits, especially in herds with Automatic Milking Systems (AMS) [9]. Rupp and Boichard [10] found that very fast milking cows tend to produce elevated somatic cell counts in milk, indicating health problems such as the occurrence of subclinical or clinical mastitis. Cows should be selected for a moderate milking speed.

Heritabilities for MT published in the literature typically range from 0.05 to 0.12 [11,12,13]. Heritabilities for MS range from 0.05 [14] to 0.22 [15]. In 2011, the team of Sewalem et al. [13] estimated the genetic parameters of MS and MT in a population of Canadian Holstein-Friesian cows. The genetic correlation coefficient was 0.25 and the phenotypic correlation coefficient was 0.10 [13]. A positive genetic correlation between MS and MT proves that cows characterized by an average or calm MT during milking gave up milk faster and within a shorter time, and this time period is longer in the case of more nervous cows.

In recent years, more and more countries in the world with highly developed HF cattle breeding have included WT into their national breeding value (WH) evaluation systems and selection indexes. However, to be able to estimate WH for this group of traits, it is necessary to know the genetic parameters. Currently, no genetic parameters of WT have been estimated in Poland. However, since 2006, data concerning phenotypic evaluation of this trait group have been collected routinely as part of the performance control system conducted by breeders’ associations. Based on the collected data, it became possible to undertake studies on the estimation of genetic parameters of WT in this work. Moreover, by using an additional source of information contained in the genome, it became possible to implement a new methodology of WH estimation based on genomic prediction. Currently, there are no papers in the scientific literature on the possibility of a more in-depth estimation of genetic parameters of WT, i.e., taking into account genomic information. Therefore, the aim of this study was to estimate genetic parameters of MS and MT and phenotypic and genetic correlations between these traits by using both pedigree and genomic data.

## 2. Materials and Methods

### 2.1. Data

The study material consisted of MS and MT phenotypes from 1,045,511 Polish Holstein-Friesian cows born between 2004 and 2013. The phenotypic evaluation of WT cows was carried out between 2011 and 2015. The dataset was created from the SYMLEK system database belonging to the Polish Federation of Cattle Breeders and Dairy Farmers (Warsaw, Poland). WTs are measured in Poland by trained classifiers by using the subjective scoring method that is developed in accordance with the recommendations of the International Committee for Animal Recording (ICAR, Stockholm, Sweden) and implemented since 2006 [16]. WTs are evaluated on a scale point, MS on a scale of 1–5 points from very slow to very fast milking cows and MT on a scale of 1–3 points from slow to excitable or even aggressive (Table 1). The WT evaluation is performed once by trained classifiers for assessment on the second test milking day in the first lactation.

The basic dataset included cow numbers, dates of birth and first calving, percent of HF breed genes and phenotypic scores for WT and parental numbers. Next, the sizes of the half-sister groups and of the herd-year-season of calving (HYS) subclasses were calculated, and their distributions were made. The half-sister group was taken as the daughter cows of the same sire and the HYS subclass was defined as the group of cows calved in the same farm, year and season, with the summer calving season covering April to September and the winter calving season covering October to March. A total of 85,172 HYS subclasses and 16,720 half-sibling groups were created. Approximately 40% of the study population consisted of cows with at least 10 half-sisters.

The distribution of half-sibling groups and HYS subclasses was presented in the Table 2 and Table 3.

Next, the date of phenotypic evaluation of workability traits was added to the basic dataset. Age at first calving was calculated from birth and calving dates. Calving dates and dates of phenotypic evaluations were used to calculate the day of lactation when the evaluation of WT was performed. Based on the day of evaluation, the cows were assigned to eight 10 day lactation stages covering the period from day 21 to day 100 of lactation. The next step was to reduce the size of the basic dataset by introducing the following restrictions: Cows with phenotypic evaluation of WT traits made later than the 100th day of lactation were removed, and WT evaluations made during trial milking performed earlier than 26 days after the first trial milking were discarded. Cows from farms where no variation in MS or MT was observed were also removed. The dataset prepared in this manner (Dataset A) counted 391,615 cows.

Based on the data contained in the genotype database of the National Research Institute of Animal Production, a dataset was created containing the genotypes of 2228 bulls whose daughters were evaluated in Poland for WT (Dataset G1). Each bull was genotyped using the Illumina BovineSNPv2 BeadChip, 54609 SNP, and genotype data were stored in the cSNP database system. Next, the imputation findhap software was used to generate genotypes dataset [17]. As a final dataset, we used 43772 SNPs with MAF > 0.001.

### 2.2. Estimation of Genetic Parameters between MS and MT

A multitrait linear model of observation including the following fixed and random effects was used to estimate the genetic and phenotypic (co)variance components of WT:*y = Xβ + Za + e*(1)
where *y* is a vector of observed traits MS/TM; *X* is the incidence matrix of fixed effects; *β* is a vector of fixed effects (HYS, lactation stage, regressions on percent of HF genes and regression on calving age); *Z* is the incidence matrix of random effects; *a* is a vector of additive genetic random effects; and *e* is a vector of random error effects.

In advance, for the estimation of the genetic and phenotypic (co)variance components of WT traits, additional restrictions were imposed on dataset A. Only the daughters of cows of 10% randomly selected sires were left in the dataset, and HYS subclasses consisting of less than 10 cows were removed. This dataset (Dataset B) of 13,280 cows was treated as the basic dataset for estimating (co)variance components of genetic and phenotypic WT traits. Then, a smaller dataset (Dataset G2) was created, containing the genotypes of 319 sire bulls from Dataset G1, which were sires from dataset B. The (co)variance components were estimated by using programs implemented in the BLUPF90 package [18] that are freely available for research purposes. The programs used were Renumf90 and Gibbs2f90 with the Bayesian Method implemented [19]. The convergence diagnostic and analysis of the posteriori distribution was conducted using the postgibbsf90 procedure included in Blupf90 package, developed by Prof. Misztal’s team at the University of Georgia (Georgia, GA, USA) During the estimation of the (co)variance components, 100,000 parameter samples were generated using a Bayesian approach via Gibbs sampling algorithm. The first 20,000 samples were removed as so-called burn-in samples, and then every fifth sample was recorded. From the subsequent samples, the heritabilities of WT traits and the genetic and phenotypic correlation coefficients between them were calculated. All the steps involved in estimating genetic parameters were repeated, taking into account in the calculations conventional data and standard pedigree information enriched with information contained in the collection with genotypes of sires (Dataset G2). The above estimations were carried out based on methodology of single-step genomic BLUP (ssGBLUP), which is one of the most frequently used methods for incorporating genomic information into genetic evaluations in livestock [20,21]. The ssGBLUP permits the use of all animals in the database by combining pedigree, genomic information from genotyped animals (usually a small portion of animals in the pedigree) and phenotypes to estimate genomic breeding values (GEBV).

Genetic and phenotypic parameters were calculated using variance components estimated with 2 different relationship matrices: pedigree-based A (BLUP) and realized H (ssGBLUP). The inverse of matrix H that combines pedigree and genomic relationships was constructed according to Aguilar et al. [21]. Cow and bull pedigrees contained two generations of ancestors, i.e., parents and grandparents, in both approaches.

## 3. Results

### 3.1. Phenotypic Characteristics of Workability Traits

The first stage of the study was to perform a detailed characteristic of the data, including the calculation of means and standard deviations and the percentage distribution of each grade for both WT by year of birth. The results of this stage of the work are presented in Table 4.

The average scores of both WT were close to the optimum in intermediate values: these values were 3.05 points for MS and 1.97 points for MT. In the following years of the birth of cows, two trends can be observed. Between 2004 and 2008, the average scores in terms of both WT were decreasing, after which the trend changed to an increasing one. Until the last analysed year of birth (2013), the phenotypic scores were increasing quite regularly, finally slightly exceeding the optimal value for each trait. The value of the coefficient of variation CV (about 20%) indicates that the right arm of the distribution of phenotypic scores in terms of WT descends more smoothly than the left one.

### 3.2. Characteristics of Polish Sire Population

The characteristics of the Polish population of PHF sires, depending on the country of origin and the method of import to Poland, are presented in Table 5. The total number of bulls included in the study was 29,578, of which 10,260 were males of domestic origin and the rest came from abroad. Among the bulls imported to Poland, the highest number came from Germany (8430) and the Netherlands (4530), and the lowest number came from France (1507) and Canada (458).

### 3.3. Genetic Parameters of WT Estimated from Pedigree Data with and without Genomic Data

The genetic parameters of workability traits estimated based on pedigree data and pedigree and genomic data of cows are presented in Table 6.

First stage was to estimate parameters based on pedigree data. In the next stage of the study, the data used for estimating the values of WT genetic parameters were enriched with information derived from the genotypes of sires. The genetic parameters of WT were estimated using information enriched from the genotypes of sires. The genotypes of 319 bulls collected in the database of the National Research Institute (Dataset G2) were included in the calculations.

Heritability estimated with genomic information was 0.11 for MS and 0.09 for MT. Next, the estimated genetic and phenotypic correlation coefficients between these traits were 0.07 and 0.12, respectively. The values of these parameters are slightly different compared to the parameters estimated without using genomic information on sire bulls. The differences between heritabilities are 0.01 and between correlations are 0.02.

The descriptive statistics (mean and Monte Carlo error) and highest posterior density (HPD) regions of the a posteriori distribution for all co(variance) estimates are presented in Table 7.

## 4. Discussion

Knowledge of genetic and phenotypic parameter, i.e., heritability and coefficients of genetic and phenotypic correlations between traits enables the optimization of selection and increases the accuracy of the prediction of its results. Typically, genetic and phenotypic correlations are used to predict changes in the value of one trait during improvement of another trait or, when the values of one trait are difficult to measure, it can be inferred from its relationship with another trait that is easier to determine.

The accurate estimation of genetic parameters of functional traits such as WT, for example, has become possible due to the increase in computational capabilities of computers, mathematical method development and increasingly sophisticated genetic parameter estimation software. The (co)variance components of functional traits were estimated by using multivariate linear observation models [22]. The authors of this paper did the same by performing genetic parameter estimation using the latest version of software developed by Prof. Misztal’s team [18]. In this work, the (co)variance components were estimated using the Bayesian method via the Gibbs sampling algorithm [23,24]. The Gibbs sampling is a method for calculating a complex posterior distribution as a steady state measure of a Markov chain. One of the problems of inference from Markov chain generation is that there will always be areas of the target distribution that have not been covered by the finite chain. Therefore, when using this method, it is important to verify the convergence and to assess the posterior distributions [25].

In our study, the convergence diagnosis (results not shown) was analyzed by using the Geweke method [26], using the algorithm implemented on the above-mentioned software. As a result, it was found that convergence was achieved for all parameter estimates because the obtained values of the convergence diagnostic test were less than one [18]. Presented in our work, the highest posterior density (HPD) region provides the interval that includes 95% of samples and is a measure of reliability. Moreover, the HPD can be applied to nonsymmetric distributions [27]. For all estimates of variance components, the HDP intervals did not include zero. Nevertheless, most of phenotypic covariance estimates showed different results, and the lower limit of intervals was less than zero. Generally, the slightly larger HDP regions for the (co)variance components were found for the estimates obtained from the genomic approach (pedigree and genomic data) than from the conventional approach (pedigree data), which suggests a slightly higher accuracy of parameter estimates when using conventional data

In this study, a two-trait animal model based on a linear model was used for the observations including both WT. The coefficient of heritability estimated in this work for MS was 0.12 (±0.0067). This result is similar to the results obtained in the study of Sewalem et al. [13], who obtained a heritability coefficient for MS equal to 0.14. A similar value for this parameter, 0.15, was obtained by Boettcher et al. [28] and Zwald et al. [29] obtained a slightly lower value of 0.11. In other works, higher heritability estimates for MS were obtained. Meyer and Burnside [14] obtained a heritability of 0.21 for the Canadian Holstein-Friesian population, while Potočnik et al. [30] estimated heritability at 0.25 by using a univariate model. The heritability of MS for Brown Swiss cattle estimated by Wiggans et al. [31] was 0.22. In a similar study to the one conducted in this paper, an Italian team of researchers performed the estimation of genetic parameters and breeding values in a population of Simmental cows [32]. The estimated heritability for milkability (very similar trait to MS) was 0.12 (± 0.01). The result of this analysis showed that genomic information could improve the accuracy of breeding values

Similar values of heritability coefficients obtained in our study may be a result of the population structure of bulls in Poland: As much as two-thirds of the population consisted of bulls imported from different countries, with the vast majority from world leading countries of cattle breeding. Such a population structure shows that the genetic variability in Poland is similar to that in the whole population of HF cattle.

Rensing and Ruten [33] estimated the genetic parameters of workability traits for Holstein-Friesian cattle by using an objective method of collecting information from milking equipment capable of measuring cow milking time. The MS heritability coefficient estimated in this manner was even higher at 0.28. The heritability of this trait, estimated based on different multivariate models applied to the first three lactations using data measured by MS recording devices in the German Simmental breed population, ranged from 0.21 to 0.40 [34]. On the other hand, a similar heritability to that published in the previously cited works, i.e., estimated using phenotypic data from electronic devices in a Hungarian Holstein-Friesian breed population, was estimated by Amin [35]; the heritability coefficient obtained by the author was 0.20. An extension of the data collected from AMS is the study by Kliś et al. [36]. The authors found that longer milking duration along with simultaneous feed consumption in the milking robot had a beneficial effect on MT. The cows were calmer and milked more easily, which also influenced the increase in milk yield.

Sewalem et al. [13] concluded from their study that the coefficient estimates may vary due to the phenotypic assessment method and analytical methods used. In addition, these authors showed that the range of heritability estimates obtained in different populations may be smaller provided that more objective methods of phenotypic evaluation of MS are used. The definition of MS in the national cattle performance monitoring program is based on the subjective evaluations of classifiers, and as a result the heritability coefficients for this trait obtained in this study are probably smaller than the results of heritability estimates published by Rensing and Ruten [30] and Dodenhoff and Emmerling [31].

The heritability 0.08 estimated from this work for MT was lower than that of MS, and this result is consistent with those published in the literature. The same exact heritability for MT was presented by Sewalem et al. [12]. A slightly lower heritability coefficient 0.04 was presented by Kramer et al. [37]. By contrast, a slightly higher heritability coefficient of MT estimated by Sewalem et al. [13] was equal to 0.13 when using the single-coefficient model and 0.20 when using the two-trait model. A similar heritability MT 0.22 for an Australian population of HF cows was reported by Visscher and Goddard [15].

The inclusion of bull genotype data had little effect on the value of heritability estimates for MS and MT; however, slightly higher estimates of genetic and phenotypic correlation coefficients between MS and MT were obtained. It is worth noting that the genetic correlations between these traits are small but positive, meaning that if the daughters of sires are milking faster, they are also generally slightly more excitable. The positive genetic correlation 0.247 (±0.075) and phenotypic correlation 0.10 between MT and MS were estimated by Sewalem et al. [13]. Wethal et al. [38] estimated genetic correlations between milking speed, temperament and leakage within milking system. The correlations were slightly higher in AMS for all combinations of traits and estimates were larger than standard errors. The genetic correlations showed absolute values ranging from 0.15 to 0.88. The genetic correlation estimated in the system milking parlour, such as in our work, was 0.16 (±0.03).

The small changes in genetic parameters after WT after the enrichment of pedigree data with genomic information can probably be explained by their structure. In this study, data including phenotypic values of cows and genotypes of their sires were used. The results of the analysis of cow genotypes were not available; consequently, there was a large disproportion between the amount of information coming from both sources. This data structure is not conducive for a high accuracy in genetic parameter estimates. In their study, Dehnav et al. [39] found that a negligible amount of information on genotyped females can improve data structure and genetic parameter estimates. Similar conclusions were presented by Cesarini et al. [32]. Furthermore, only cows with their own phenotypes and sires with a relatively large number of daughters with phenotypes were used in the calculations. These conditions were probably the main reasons for the very similar estimates of the WT genetic parameters that were obtained with the use of pedigree data and of pedigree and genomics.

In practice, low coefficients of heritability of a trait indicate the possibility of its effective genetic improvement in a selected animal population (direct selection), but can simultaneously mean a slower genetic gain than in the case of highly heritable traits. One solution is the indirect selection for other moderate and highly heritable traits that are strongly and positively genetically correlated with low heritable traits. Therefore, the continuation of these studies should be the estimation of genetic and phenotypic correlation coefficients between WT traits and other genetically improved production and functional traits.

## 5. Conclusions

In conclusion, the obtained coefficients of heritability and genetic and phenotypic correlations between WT indicate a possibility of effective improvement of the population in terms of this group of traits.

The analysis of the values of coefficients of genetic parameters of WT estimated on the basis of pedigree and genomic data showed that the enrichment with information derived from the genotype of bulls did not bring significant changes in the values of the heritability and correlation coefficients. However, further studies with larger data sets included genotypes of cows are recommended.

## Figures and Tables

**Table 1 animals-11-02443-t001:** Workability traits scale points.

Category	Milking Speed (MS)	Milking Temperament (MT)
1	Very slow milking speed	Slowly animal
2	Slow milking speed	Normally responsive animal
3	Average milking speed	An excitable or aggressive animal
4	Fast milking speed	
5	Very fast milking speed	

**Table 2 animals-11-02443-t002:** Distribution of half-sibling groups.

Size of Half-Siblings Group	Number of Half-Siblings Groups
1	4690
2–5	3600
6–9	1575
10–19	1518
20–49	1543
50–99	1498
100–199	1371
200–499	558
500–999	213
1000–1999	106
2000–4999	39
5000–9999	8
10,453	1
Total	16,720

**Table 3 animals-11-02443-t003:** Distribution of HYS groups.

Size of HYS Group	Number of HYS Groups
1	21,756
2–5	38,429
6–9	11,642
10–19	8528
20–49	3271
50–99	1223
100–199	301
200–499	22
Total	85,172

**Table 4 animals-11-02443-t004:** Number (N), means ($\bar{x}$) and standard deviations (SD) of workability traits in Polish Holstein-Friesian cattle population by the year of birth.

Year of Birth	N	MS			MT		
		$\bar{x}$	SD	CV (%)	$\bar{x}$	SD	CV (%)
2004	97,702	3.02	0.68	22.52	2.00	0.37	18.50
2005	105,877	3.03	0.70	23.10	1.98	0.37	18.69
2006	107,927	3.03	0.74	24.42	1.96	0.38	19.39
2007	108,394	2.98	0.81	27.18	1.93	0.41	21.24
2008	120,026	2.92	0.86	29.45	1.90	0.43	22.63
2009	121,766	2.96	0.84	28.38	1.91	0.42	21.99
2010	118,231	3.11	0.66	21.22	1.98	0.33	16.67
2011	127,019	3.15	0.61	19.37	2.00	0.30	15.00
2012	121,313	3.18	0.59	18.55	2.01	0.27	13.43
2013	17,256	3.19	0.61	19.12	2.02	0.26	12.87
Total	1,045,511	3.05	0.73	23.33	1.97	0.37	18.04

**Table 5 animals-11-02443-t005:** Characteristics of populations of Polish Holstein-Friesian (PHF) sires used in Poland depending on the country of origin and method of import to Poland.

Country	Import Type				Total
	National	Semen	Sire of Heifer	Other	
Germany		747	7418	265	8430
The Netherlands		657	3832	41	4530
United States		781	650	250	1681
France		241	714	552	1507
Denmark		119	1023	12	1154
Canada		249	150	59	458
Other countries		425	1110	23	1558
Poland	10,260				10,260
Total	10,260	3219	14,897	295	29,578

**Table 6 animals-11-02443-t006:** Genetic and phenotypic parameters of workability traits based on conventional data and conventional and genomic data.

Trait	MS	MT
**pedigree data**		
h^2^	0.12 (±0.0067)	0.08 (±0.0063)
r_gMS,MT_	0.05 (±0.0002)	
r_pMS,MT_	0.14 (±0.0004)	
**pedigree and genomic data**		
h^2^	0.11 (±0.0065)	0.09 (±0.0012)
r_gMS,MT_	0.07 (±0.0003)	
r_pMS,MT_	0.12 (±0.0005)	

h^2^—heritability; r_g_—genetic correlation; r_p_—phenotypic correlation; MS—milking speed; MT—milking temperament.

**Table 7 animals-11-02443-t007:** Posterior means, Monte Carlo error (MCE) and highest posterior density (HPD) region of genetic and phenotypic parameter estimations of workability traits based on conventional data and conventional and genomic data.

Parameter	Mean	MCE	HPD Interval 95%	
**Pedigree data**			**Low limit**	**High limit**
Genetic (co)variance				
σ^2^_MS_	0.02588	0.0038	0.00031	0.00472
cov	0.00055	0.0001	0.00019	0.00263
σ^2^_MT_	0.00462	0.0008	0.00033	0.00201
Phenotypic (co)variance				
σ^2^_MS_	0.21570	0.0003	0.20620	0.21910
cov	0.01560	0.0021	−0.00110	0.00395
σ^2^_MT_	0.05770	0.0001	0.05494	0.05856
**Pedigree and genomic data**				
Genetic (co)variance				
σ^2^_MS_	0.02373	0.0125	0.00008	0.00564
cov	0.00078	0.0004	−0.00110	0.00147
σ^2^_MT_	0.00518	0.0003	0.00035	0.00146
Phenotypic (co)variance				
σ^2^_MS_	0.21570	0.0008	0.20690	0.22060
cov	0.01340	0.0012	−0.00034	0.00461
σ^2^_MT_	0.05760	0.0001	0.05509	0.05853

σ^2^—variance; cov—covariance; MS—milking speed; MT—milking temperament.

## Data Availability

The data presented in this study came from two sources: 1. Pedigree and phenotypic data came from the database system SYMLEK, belonging to the Polish Federation of Cattle Breeders and Dairy Farmers, Warsaw, Poland which is available for research purposes at no charge, 2. Genomic data came from internal (National Research Institute of Animal Production, Cracow) database system which is also available in Poland for research purposes.

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
