# Peer review of "Genetic Parameters of Workability Traits in the Population of Polish Holstein-Friesian Cows Based on Conventional and Genomic Data"

_animals, 2021, doi:10.3390/ani11082443_

Round 1

Reviewer 1 Report

Dear authors,

The aim of the manuscript is of interest and the paper is overall well-done. However, some important details are missing, and some points must be addressed before to consider it ready for publication. Just two examples: i) you need to include standard deviations for heritabilities and genetic correlation to give more information about their reliability; ii) you need to specify what kind of genotypes you had and how you used them in the model. Discussion is coherent and abundant, but I suggest you a manuscript that is quite similar to yours 10.3168/jds.2020-19838; please consider using it for some thoughts. Please see the line-by-line comments listed below. M&M section must be improved, as well as the result section.

Line-by-line

Lines 31-32: these heritabilities were estimated using pedigree + genomics, right? If so, please specify

Lines 58-59: “The genetic correlation coefficient was 0.25 and the phenotypic correlation coefficient was 0.10.” are the values coming from the reference 13?

Lines 69-71: What does “… additional source of information contained in the genome…” exactly mean? Did you have “normal” genotypes (i.e., SNPs) for males?

Line 83: please fix “¬–“

Lines 86-105: the description of the dataset is quite hard to follow. For example, there is no need to say how you computed the age at first calving or the day of lactation, their computation is already in their name.

Lines 106-107: please include details about the genotypes: which beadchip? How many SNPs? Imputation? Quality control? And so on…

Lines 119-122: please justify the reasons of using a smaller dataset to estimate VC

Lines 125: which “gibbsf90” software did you use? Gibbs1f90, gibbs2f90, or gibbs3f90?

Lines 126-127: did you save all samples after the burn-in?

Lines 129-132: it means that you ran a ssGBLUP? If so, please include references (e.g., Legarra et al., 2009; Aguilar et al., 2010). If not, please specify how did you consider the genotypes.

Lines 134-136: please remove these lines coming from the template

Lines 147-149: please explain this statement. I do not know how CV can be related to the left or right part of the distribution, which is generally studied through skewness.

Line 155, Table 2: please add standard deviations for heritabilities and genetic correlation. Was the phenotypic correlation significant?

Line 165, Table 3: in which way this table is related to the aim of the work? You can consider reporting MS and MT values according to the country of origin of sires.

Line 173: “Heritability of MS was 0.11 and MT was 0.09, was presented in table 4.” Please rephrase this sentence.

Lines 176-177: correlations are 2 points lower compared to the values in Table 2.

Line 178, Table 4: please add standard deviations for heritabilities and genetic correlation. Was the phenotypic correlation significant?

Lines 181-182: I would not consider phenotypic correlation among the genetic parameters.

Lines 183-187: again, I do not strongly agree that phenotypic correlation can be used to infer a phenotype based on another; this is done using the genetic correlation.

Line 197: remove the brackets () and add the SD

Best regards

Author Response

Dear Reviewer,

We would like to thank you very much for your review. The suggestions from your review are very valuable for improving the paper therefore we have incorporated them into the current version of the manuscript. Thank you for your general feedback about i) standard deviations and ii) kind of genotypes, we have added them to manuscript.

Line-by-line:

  1. Lines 31 – 32: It was specified and include in the current version of the paper.
  2. Lines 58 – 59: Yes, those values came from reference 13. We have added that to text.
  3. Lines 69 – 71: „… additional source of information contained in the genome…” means, that information came not only from pedigree and phenotype but also from genotype.
  4. Line 83: We fixed this sign in the manuscript
  5. Lines 86 – 105: In the manuscript, we have described the whole creation process of each subset of data because we really wanted to show that process. We wanted to justify that the final reduced input datasets were derived from very large datasets by applying appropriate restrictions.
  6. Lines 106 – 107: We have included those informations in the text.
  7. Lines 119 – 122: The reason of using a smaller dataset to estimate VC was to limit the size of the main dataset for computational reasons. Lastly, 10% of the sires of cows were randomly selected. In addition, we wanted select resonably large sizes of HYS subclasses. For small frequencies, cows assigned to HYS subclasses have few or no hals-fiblings and it becomes difficult or impossible to separate genetic from environmental effect.
  8. Line 125: We have used Gibbs2f90. It has been corrected in the manuscript.
  9. Lines 126 – 127: We have saved and recorded every fifth sample after the burn-in. We have added this to the manuscript.
  10. Lines 129 – 132: Yes, we have used ssGBLUP in our research. We include those information and references in the manuscript.
  11. Lines 134-136: Thank you for this comment. These lines were removed from the manuscript.
  12. Lines 147 – 149: When the distribution is close to normal and the CV is large (e.g., 25%) it indicates skewness (extended right arm of the distribution). It can be thought that outliers are present.
  13. Line 155: We have added and include standard deviations for heritabilities and correlations in the manuscript.
  14. Line 165: We wanted to well-present the genetic structure of the Polish sub-population of HF cattle. That population is influenced very much by the import of genetic material from other countries. We also wanted to show that the results obtained in our research can be applied to the whole metapopulation of HF cattle. This was discussed in the discussion.
  15. Line 173: This sentence was rephrased in the text.
  16. Lines 176 – 177: We have corrected that in the text.
  17. Line 178: We have added those standard deviations to the manuscript.
  18. Lines 181 – 182: Of course, we have corrected that in the text.
  19. Lines 183 – 188: Of coures, we have changed thai in the manuscript
  20. Lines 197: The brackets were removed and standard deviations were added.

Thank you once again,

Authors

Reviewer 2 Report

The paper is overall well written and easily read through. However, I found some of the par incomplete or at least not too deep as to match the relevance that the topic has. Furthermore, conclusions are vague at they do not really report any take-home message which directly stems for the outcomes of this study. Hence, I suggest authors consider the suggestions below and try to work hard on the discussion that they present. Otherwise, I feel the paper should be better considered a short communication.

Author Response

Dear Reviewer,

We would like to thank you very much for your review.

Unfortunately, we cannot see your 'suggestions bellow'. The current version of the paper has been much improved, according to your general feedback. Which was mostly in line with the detailed suggestions of the other reviews. Therefore we hope that the current version of the paper is acceptable.

Thank you once again,

Authors

Reviewer 3 Report

The manuscript “Genetic parameters of workability traits in the population of Polish Holstein-Friesian cows based on conventional and genomic data” presents heritability and correlation estimations for milking speed (MS) and temperament (MT) based on pedigree and combined pedigree and genomic data. The topic is of interest to the field, considering the importance of functional traits to production costs and their recent inclusion in selection indices. However, the manuscript should be improved. The authors should provide more details in the methodology section and the presentation of the results can be better organized. However, my main concern is related to the model used for analyses. The linear model assumes normal distributed data. Nevertheless, the study involved categorical traits. This can be an issue especially for MT, for which the phenotypes are only three categories. Therefore, it is important for the authors to present the data distribution graphically as well as to perform convergence diagnostics. A suggestion is to use threshold models for such categorical variables. 

Other important issues are: 

  1. The authors aimed to compare the use of pedigree and pedigree plus genomic data in genetic parameter estimation but these approaches were only compared regarding genetic parameter values. No comparison regarding precision and accuracy of the estimates was made. Provide measures as HPD and Monte Carlo error to discuss about that either. Moreover, include covariance estimates (and not only genetic parameter estimates)
  2. The genotype data is not described in the manuscript (for example, how many SNP were considered, quality control, etc.)
  3. The models regarding the two-trait analyses as well as the model including genomic data were not described in the methodology. Also, add information regarding pedigree data. 
  4.  The results regarding both approaches can be presented together, in the same section. Section 3.3 should be presented before genetic parameters and after section 3.1, since it is a description of the population. 

Other minor comments are presented below: 

Line 14: It is better to use “considering two approaches” than that “in two stages”. The latter gives the idea that it is one analysis divided into two steps and not two different analyses. 

Line 16: Indicate as a low correlation (almost null). Also, the traits are low heritable and not moderately heritable. 

In Abstract, it is mentioned that the dataset is formed by phenotypic evaluations of 1,045,511 cows. However, the number of records actually used in estimation is much lower. Perhaps it is better to include the description of the actual (and final) dataset used. 

Line 36: It is redundant to use “workability traits” and “Holstein” as keywords since they already appear in the title. 

Line 78: As a suggestion, to avoid the excessive use of “phenotypic evaluations” in the paragraph, rephrased it to “The study material consisted of MS and MT phenotypes from 1,045,511 Polish Holstein-Friesian cows born between 2004 and 2013”. 

Line 92: Inform the minimum and maximum number of cows belonging to HYS subclass and half-sibling group. 

Line 101: Why the authors restricted the data to evaluations between 21 and 100 days of lactation?

Line 100-110: It appears that only Dataset B and G2 were used in the analyses. Therefore, the authors should describe only these datasets (no need to describe Dataset A and G1). Also, include the description of Dataset B in this section as well. 

Line 127: Bootstrap samples or burn-in? No thin value was used?

Line 129: Did the authors evaluate the convergence? How the number of iterations was defined?

Line 139: “The percentage distribution of each grade by year of birth” is the coefficient of variance? If yes, use the correct term (coefficient of variation). 

Line 234: single-coefficient = single-trait?

The authors discussed the heritabilities comparing them with the values obtained in other studies. However, were these low heritability values expected? What is the impact on selection/animal breeding programs that aim to genetically improve such traits?

Line 238-242: Such correlation values were expected? Explore more the low and positive correlation between the traits. 

Line 243: It is not clear the authors’ hypothesis for the small differences between the two approaches regarding heritability and correlation estimates. Other studies which compared both approaches in genetic parameter estimates were not used in the discussion. 

Author Response

Dear Reviewer,

We would like to thank you very much for your review.

The suggestions from your review are very valuable for improving the paper therefore we have incorporated them into the current version of the manuscript.

Regarding your comment about the kind of observation model we have used, we wanted to explain the genesis of using a linear model:

To the analysis of traits which have a categorical distribution, taking two or more values, two groups of methods can be used. The first one assumes that the trait has normal distribution and can be described by a linear model. The second group of those methods based on the so-called threshold model. This models assumess that underlying the variability of the trait with a categorical distribution lies an underlying variable with a continuous distribution. The equations of the threshold model are not linear, and solving their system is very complicated and requires much more computational power than the linear model.  Theoretically, threshold models better account for the probalistic nature of traits with a categorical distribution, but in practice many authors using both models have not confirmed the superiority of the threshold model over the linear model when estimating genetic parameters (Hager and Hofer, 1989; Jamrozik et all, 1991, Varona et all, 1999). Nowadays, most of the scientific research on the estimation of (co)variance components of workability traits are based on the method including linear models (Sewalem at all, 2011; Casarani et all, 2020). Therefore, we decided to choose this model to our research. The usefulness of the threshold model for estimating VC components will be investigated in further studies on this issue.

Thank you for the other important issues:

  1. In current version of manuscript we have added standard deviations and (co)varinace components.
  2. Informations about genotypes were added to the text
  3. Those informations have been added in the mauscript.
  4. The order of the sections has been changed in the text.

Other minor comments :

  1. Line14: It was corrected in the text.
  2. Line 16: We changed that in the manuscript.
  3. Line 36: Thank you for that suggestion, we rearanged the kewords section.
  4. Line 78: This sentence was rephrased in the manuscript.
  5. Line 92: We added two new tables that show HYS subclass and half siblings groups.
  6. Line 101: These phases were chosen because the phenotypic evaluation of milking ability traits in Poland is performed on day 2 of the trial milking.
  7. Lines 100 – 110: The reason of using a smaller dataset to estimate VC was to limit the size of the basic dataset , of course. Additionally, 10% of the sires of cows were randomly selected. In addition, we wanted to select reasonably large sizes of HYS subclasses. For small frequencies, cows assigned to HYS subclasses have few or no siblings and it becomes difficult or impossible to separate genetic from environmental effect. We wanted to justify that the final reduced input datasets were derived from very large datasets by applying appropriate restrictions.
  8. Lines 127, 129: It has been described in the manuscript.
  9. Line 139: It was changed in the text. We meant the number not the percentage distribution
  10. Line 234: Of course, we meant trait. We changed this in te manuscript.
  11. Suggestion about the impact of those heritabilities on selection has been explained in the text. Lines 285 - 295.
  12. Lines 238 – 242: That was explored more in the manuscript.
  13. Lines 243: This information has been changed in the text.

Thank you once again,

Authors

Round 2

Reviewer 1 Report

Dear authors,

the manuscript is now improved. I found just few minor errors, please see the comments below.

Best wishes

Line-by-line comments:

Line 27: "391,615" should be "391,615 cows"?

Line 28: 12280 is not the 10% of 391615. Where is this number from?

Line 203: "The results were presented in the table 6" should be "The results are presented in the Table 6" because the other sentences about tables are in the present form.

Line 238: "... and breeding value..." should be "... and breeding values..."

Line 284: ". correlations were ..." should be with the capital letter ". Correlations were ..."

Lines 286-287: please remove the duplicated word Genetic.

Lines 286-287: "... like in our work, 0.16..." should be "... like in our work, was 0.16 ..."

Author Response

Dear Reviewer,

We would like to thank you very much for your review. Thanks also for pointing out minor errors.

Line-by-line:

  1. Line 27: "391,615" should be "391,615 cows"?

Line 27: It was changed in the current version of manuscript.

  1. Line 30: 12280 is not the 10% of 391615. Where is this number from?

Line 28: We have corrected that issue in the manuscript. There was a mistake. We have selected randomly not 10% of the cows, but 10% of the sires which were fathers of 13,280 cows. That was also basic dataset to estimate genetic parameters.

  1. Line 203 "The results were presented in the table 6" should be "The results are presented in the Table 6" because the other sentences about tables are in the present form.

Line 214: We have rewritten sentence and changed tables.

  1. Line 238: "... and breeding value..." should be "... and breeding values..."

Line 298: We have changed the form to plural

  1. Line 284: ". correlations were ..." should be with the capital letter". Correlations were ..."

Line 345: The sentence starts now from capital letter.

  1. Lines 286-287: please remove the duplicated word Genetic

Line 347: Thank you for that. It has been removed

  1. Lines 286-287 like in our work, 0.16..." should be "... like in our work, was 0.16 ..."

Lines 348: We have changed that sentence.

Thank you once again, 

Authors

Reviewer 2 Report

I am sorry as I enclosed comments to editor but the authors. Please find my previous review enclosed.

The paper is overall well written and easily read through. However, I found some of the par incomplete or at least not too deep as to match the relevance that the topic has. Furthermore, conclusions are vague at they do not really report any take-home message which directly stems for the outcomes of this study. Hence, I suggest authors consider the suggestions below and try to work hard on the discussion that they present. Otherwise, I feel the paper should be better considered a short communication.

Remove: Lines 133 to 136 (already removed. Do not pay attention to this)

Line 198, 201, 202, 203, 212, 218. Remove parentheses.

Milking behavior must be better described. If I did not understand wrong different levels within the variable may represent behavioural patterns on a scale. This is complex as the scale reported may comprise behaviorual patterns of a very different nature. For instance, and even if it is already published somewhere else, when the authors report cows to be slow to excitable or even aggressive in a scale from 1¬–3, this scale is reporting behavioural patterns which are not even opposites that can be included in the same scale. Often it is just a matter of scale definition, but still authors need to provide much more information in regards how this was implemented to ensure the paper can be replicable.

I enjoyed the discussion but I find it a little bit superfluous and unfocused at some points. By this I mean, the authors have an important database and that is ok, still the information that can be taken out of the methods applied was not deeply discussed at some points, which also translated in an 8 pages long paper with everything included (when Bayesian and frequentist approaches has been described). More information must be presented in regards Bayesian approaches, although it is true that referencing saves much space and helps paper not becoming everlastingly long, when it comes to Bayesian approaches, you cannot just provide a reference to a manual or a technique and then leave it for author discretion to whether consult or not the resources that authors used, as Bayesian statistics involve which algorithms were used, which priors were used, how posterior distributions were assessed, among others.

Of course, I feel the paper if the aforementioned comments are approached enjoys of sufficient entity as to be published. However, in its present form I am not clear on whether this should be better considered a short communication. Let’s wait for a second round to be able to decide.

Author Response

Dear Reviewer,

We would like to thank you very much for your review. Thanks also for pointing out minor errors.

Milking behavior must be better described. If I did not understand wrong different levels within the variable may represent behavioural patterns on a scale. This is complex as the scale reported may comprise behaviorual patterns of a very different nature. For instance, and even if it is already published somewhere else, when the authors report cows to be slow to excitable or even aggressive in a scale from 1¬–3, this scale is reporting behavioural patterns which are not even opposites that can be included in the same scale. Often it is just a matter of scale definition, but still authors need to provide much more information in regards how this was implemented to ensure the paper can be replicable.

WT are measured in Poland by trained classifiers using the subjective scoring method. This method has been developed in accordance with the recommendations of the International Committee for Animal Recording and implemented in Poland since 2006. Classifiers are regularly trained and several times a year during meetings they evaluate the same cows in terms of WT. This is to compare the results of these measurements to ensure the greatest possible objectivity of the scoring (they check what is the scale of divergence/ integrity in the evaluation of the same animals and correct them). In the current version of the manuscript we have provided more information on this phenotypic evaluation and on the scale of measurements in WT, but you know, we are not specialists. We also added a table containing scoring scales and behavior definitions. Table 1 – Line 101. 

I enjoyed the discussion but I find it a little bit superfluous and unfocused at some points. By this I mean, the authors have an important database and that is ok, still the information that can be taken out of the methods applied was not deeply discussed at some points, which also translated in an 8 pages long paper with everything included (when Bayesian and frequentist approaches has been described). More information must be presented in regards Bayesian approaches, although it is true that referencing saves much space and helps paper not becoming everlastingly long, when it comes to Bayesian approaches, you cannot just provide a reference to a manual or a technique and then leave it for author discretion to whether consult or not the resources that authors used, as Bayesian statistics involve which algorithms were used, which priors were used, how posterior distributions were assessed, among others.

According to your suggestions, we've added more information about the Bayesian method in the current manuscript version. We added information about the algorithm used, the assessment of convergence and posterior distribution. In this regard, we have improved the material and methods, results and discussions section. Line 269.

Line-by-line:

Line 198, 201, 202, 203, 212, 218. All parentheses has been removed.

Thank you once again, 

Authors

Reviewer 3 Report

The manuscript improved after the revision. However, some questions were not fully addressed by them. The details are given below:

The inclusion of the standard deviations was valuable. However, it gives the idea of variability and not of precision or accuracy. Please include the HPD 95% as well. It would help the comparison of the approaches. 

Line 24: Change "were used" with "were available" 

Line 31: Change to "Heritabilities (standard deviation)" and include the standard deviations between brackets. The plus/minus sign gives the idea of credibility interval (which is not). 

Line 124: Assign a reference to findhap software. 

Line 152: The method used to incorporate genomic information was described; however, it would be valuable to include that the difference between this and conventional analysis is related to the relationship matrix used. In addition, the model presented is related to a single-trait analysis. Please, include the model for two-trait analysis as well. 

It is still not clear if chain convergence was evaluated. 

The authors mentioned in the review that they included the variance component estimates in the manuscript; however, this was not done. 

Information from tables 5 and 6 can be presented in one single table and both sections can be unified in only one section, named "Genetic parameters for WT", for example. 

Line 202: This information was already given in the previous paragraph. 

Line 203-206: The information from the table is given throughout the text as well (redundancy).

Line 208: Replace "are" with "were". 

Line 281-288: This information was not adequately linked to the previous information.   

Recent studies that compared polygenic and genomic-polygenic models in genetic parameter estimation were not used in the discussion.

Author Response

Dear Reviewer,

We would like to thank you very much for your review. Thanks also for pointing out minor errors.

The inclusion of the standard deviations was valuable. However, it gives the idea of variability and not of precision or accuracy. Please include the HPD 95% as well. It would help the comparison of the approaches.

HPD 95% it has been included in table 7 (line 234)

Line-by-line:

Line 24: Change "were used" with "were available"

Line 25: It was changed in manuscript.

Line 31: Change to "Heritabilities (standard deviation)" and include the standard deviations between brackets. The plus/minus sign gives the idea of credibility interval (which is not).

Line 36: In current form of the manuscript all of standard deviations are between brackets. Thank you for this.

Line 124: Assign a reference to findhap software

Line 135: We have assigned reference to findhap software.

Line 152: The method used to incorporate genomic informationwas described; however, it would be valuable to include that thedifference between this and conventional analysis is related tothe relationship matrix used. In addition, the model presented isrelated to a single-trait analysis. Please, include the model fortwo-trait analysis as well. The model was changed in current form of paper.

Line 162 Convergence chain The description of the convergence chain was extended in the paper

The authors mentioned in the review that they included the variance component estimates in the manuscript; however, this was not done.

Line 234: Sorry for not including the variance component estimates. Now they are presented in Table 7.

Information from tables 5 and 6 can be presented in one singletable and both sections can be unified in only one section, named "Genetic parameters for WT", for example.

Those tables and section have been changed. It is now Table 6 – line 229

Line 202: This information was already given in the previous paragraph.

That paragraph has been changed in current form of paper.

Line 203 – 206 The information from the table is given through out the text as well (redundancy).

At newest version of manuscript we have tried to avoid redundancy. It has been changed.

Line 208 Replace "are" with "were".

Line 220 That word has been replaced.

Line 281-288 This information was not adequately linked to the previous information.

Line 338 That information has been changed and linked more adequately to the previous information

Recent studies that compared polygenic and genomic-polygenicmodels in genetic parameter estimation were not used in thediscussion.

We have added a few recent studies that compared polygenic and genomic-polygenic models for example, Cesarani et al. (2021).

Thank you once again,

Authors
